# Infant Mortality Trends and Determinants in Kazakhstan

**DOI:** 10.3390/children10060923

**Published:** 2023-05-24

**Authors:** Nurbek Yerdessov, Olzhas Zhamantayev, Zhanerke Bolatova, Karina Nukeshtayeva, Gaukhar Kayupova, Anar Turmukhambetova

**Affiliations:** School of Public Health, Karaganda Medical University, Gogol Street 40, Karaganda 100008, Kazakhstan

**Keywords:** infant mortality, health determinants, middle-income country, Kazakhstan, Central Asia

## Abstract

Infant mortality rate (IMR) is a crucial indicator of healthcare performance and a reflection of a country’s socioeconomic development. We analyzed the trends of IMR in Central Asia (CA) countries and its determinants in Kazakhstan, which is a middle-income country. Linear regression was used for IMR trend analysis in CA countries from 2000 to 2020 and for exploring associations between IMR and socioeconomic factors, health service-related factors, and population health indicators-related factors. A gamma generalized linear model was applied to define associations with various determinants. Our analysis revealed that IMR has decreased in all CA countries, with Kazakhstan having the lowest rate in 2000 and 2020. Our results suggest that socioeconomic indicators, such as total unemployment, Gini index, current health expenditure, gross domestic product (GDP), proportion of people living in poverty, and births by 15–19-year-old mothers, were associated with increased infant mortality rates. Improving socioeconomic conditions, investing in healthcare systems, reducing poverty and income inequality, and improving access to education, are all potential issues for further development. Addressing these factors may be critical for improving maternal and child health outcomes in the region.

## 1. Introduction

Infant mortality rate (IMR) is widely considered as a major public health criterion and a reflection of a country’s socioeconomic development. It is closely linked to maternal health and is a key determinant of a nation’s health [1,2]. For countries with limited resources, infant mortality remains a relevant and easily calculable measure of population health [3]. The IMR is a standardized measure of children’s deaths under one year of age per thousand live births. Achieving a reduction in IMR is a crucial milestone in attaining Sustainable Development Goal 3, with the purpose of providing healthy and active lives for every human being. The global infant mortality rate (IMR) has shown a significant decline, from 65 infant deaths per 1000 live births in 1990 to 29 infant deaths per 1000 live births in 2018 [4].

Kazakhstan has witnessed a notable reduction in mortality rates among all age groups, including infants, since 2000 [5]. Nevertheless, despite considerable investments in the health sector, Kazakhstan continues to fall behind OECD countries when it comes to key health indicators. The population of the country, which faced a decline between 1992 and 2002, has experienced a subsequent growth of 20%. As of 2019, Kazakhstan’s gross domestic product (GDP) was estimated at USD 180.2 billion, with a per capita gross national income of USD 8810. These figures are significantly lower compared to the per capita gross national income of USD 40,115 observed in other OECD countries [6]. According to time-series analysis conducted by Thomas, the transition to a market economy in the 1990s temporarily alleviated environmental degradation but also resulted in socioeconomic challenges and a decline in living standards [7]. Some credibility issues have been raised regarding infant mortality data, including underreporting/misreporting in the Central Asian region and Eastern European countries [8,9]. However, in 2021, Kazakhstan, Kyrgyzstan, Uzbekistan, and Tajikistan were among the 36 countries with high quality national data used for child mortality estimation by UN experts [5]. Furthermore, Kazakhstan implemented international criteria for live births and stillbirths in 2008, leading to increased attention and resources for the care of low weight newborns [10]. Over the past decade, mortality rates in Kazakhstan have declined across all age groups, including infants [11]. Regions such as Kostanay, Akmola, and Almaty have observed relatively high neonatal mortality rates [12]. In the Central Asian countries, neonatal mortality has experienced a 66% decline between 1990 and 2021, with an average annual reduction of 3.4% [5].

Uzbekistan has also achieved a decline in the death rate across all age groups, thanks to improvements in the healthcare system and an enhanced focus on maternal and child health [13]. Despite its relatively low GDP level, Kyrgyzstan has made significant progress in reducing infant mortality, which was successfully reduced by 46% over the last two decades [14].

Official data provide consistent evidence that IMR has decreased steadily in the EU and its member states over the past few decades, particularly in Baltic states and some East European countries. However, some developed countries have seen an increase or plateau in IMR in the last decade [15]. In the mid-2000s, it was found that South Korea was managing infant mortality levels relatively well. The IMR in South Korea was 3.2 in 2009, which was lower than the average IMR of OECD nations (4.7 in 2008) and the USA (6.3 in 2009), but higher than Japan’s IMR (2.8 in 2009). This difference was attributed to the healthcare service system in Korea. Recommendations were made to improve perinatal, neonatal, and infant healthcare, such as establishing new policies for caring for preterm and high-risk pregnancies, and research driven perinatal facilities [16]. Meanwhile, in Japan between 1999 and 2017, the IMR differed based on household occupation types. The IMRs for farming households almost doubled (1.96), and for unemployed households, it increased 6.5 times compared to occupation types with the highest income [17]. In the USA, between 1999 and 2017, the IMR dropped from 736.0 to 567.0 per 100,000 live births, while the age of women giving birth increased [18,19]. In Chile, between 1990 and 2011, the IMR declined rapidly with an annual percent change (APC) of −5.4. However, from 2001 to 2016, the rate of decline slowed down (APC −1.6) [20]. In Scotland, between 2000 and 2018, IMR and stillbirth rates decreased. Nonetheless, socioeconomic disparities continue to exist, and there are indications that mortality rates among the disadvantaged communities might be deteriorating [21]. All the countries mentioned above belong to the group of high-income countries [22].

We also looked at the IMR situation in some middle-income countries. In Hungary, between 1963 and 2012, a notable seasonal (end of winter) impact on neonatal and infant mortality was identified. This effect was more pronounced among infants with low birthweight and mothers with lower levels of education. It is suggested that respiratory infections may be linked to this observed phenomenon [23]. In Venezuela, achievement in managing the IMR came to a halt around 2009. Despite previous success in reducing infant mortality, the IMR in Venezuela reached 21.1 deaths per 1000 live births in 2016, which was almost 1.4 times higher than the rate in 2008. This represents a significant setback for public health efforts to improve maternal and child health in the country [24]. While India has made significant strides in reducing IM over the last decades, the country still faces challenges in ensuring the survival and well-being of all children. IMR varies widely across different states and socioeconomic groups [25]. A study in China analyzing the IMR from 1999 to 2019 suggested that inflation has an impact on infant mortality [26].

### 1.1. Determinants

The World Health Organization (WHO) emphasizes that maternal and child health is influenced by various social determinants of health that extend beyond access to healthcare. In addition to health policies, addressing socioeconomic factors and living conditions is equally crucial for infant survival [27]. Various social determinants, including gender, ethnicity, socioeconomic status, urban/rural disparities, work opportunities and conditions, and the broader context (such as state regulations and policies, and culture), significantly impact infant health outcomes. For instance, economic development at the country level, measured by GDP per capita, seems to have an impact on IM to some extent during particular stages of human life [28,29]. Per capita government expenditure on healthcare emerged as the primary factor influencing infant mortality rates in Asia. Asian nations that allocated higher per capita funds to healthcare experienced notably lower levels of IMR [30]. The Gini index, which measures income distribution across a population, indicates the level of inequality present. A higher Gini index suggests greater inequality [31]. There was a positive and significant correlation observed between infant mortality and the Gini index [29]. Higher human development is strongly correlated with decreased infant mortality [32]. Additionally, income inequality is found to be positively associated with IMR, while mean household income and female educational attainment are negatively correlated with IM. The unemployment rate is found to be independently associated with IM [28,33]. In Italy, despite the availability of universal healthcare, differences in IM were linked to unemployment and income level [29].

Recent research found that adequate funding of healthcare systems is crucial. These studies demonstrated that, for every 1 percent increase in public health expenditure, there is a corresponding decrease in the IMR of approximately 0.6 to 1 percent [34,35].

Additionally, healthcare policies and actions have a significant impact on the reduction of IM. In Brazil, establishing the unified health system and the family health strategy were important factors in reducing infant mortality [33]. Other factors affecting rates of infant mortality were found to be income, poverty, nutritional status, housing, and educational attainment. Decreasing fertility rates were also associated with decreasing infant deaths [36].

Environmental factors also play a role in infant mortality rates. For instance, poor working environments and industrial pollution are significantly related to infant mortality, while urbanization, employment in the service sector, and economic wealth are not [37]. Furthermore, research indicates that domestic violence poses a risk to infant survival, as infants born to women who have encountered two or more incidents of domestic violence face increased rates of IMR [38].

### 1.2. The COVID-19 Period

Although the COVID-19 pandemic has had little direct effect on IM, its indirect impacts on the economy and the effectiveness of the health system are anticipated to result in higher death rates among this vulnerable population in low- and middle-income nations. In cases where the mother is positive for SARS-CoV-2 PCR, the baby is at increased risk of complications and adverse outcomes, which may include specialized neonatal care and prolonged hospitalization [22,39].

Previous research reported difficulties in delivering healthcare services to families, particularly in rural areas, and a disrupted referral chain, resulting in a decrease in stillbirth and infant mortality rates, during the COVID-19 pandemic. However, the pandemic has also led to a reduction in the number of follow-up visits during pregnancy, and pregnant women have had their regular antenatal care disrupted [40,41]. In addition, the lockdown policy due to the pandemic, coupled with patients’ concerns about becoming infected in hospital, may have contributed to the higher stillbirth rate [42].

Strict preventive efforts implemented by the government at the outset of the pandemic have led to a comparatively low prevalence of COVID-19 in Kazakhstan, a middle-income country in CA [43,44]. A rise in unemployment, a reduction in economic development, and aggressive inflation were all occurring at the same time. Access to healthcare has decreased despite attempts to preserve basic services [45]. Given these complex factors, our aim is to estimate infant mortality rate trends from 2000 to 2020 in Kazakhstan and CA and identify the IM determinants in Kazakhstan.

## 2. Materials and Methods

### 2.1. Data Sources

A retrospective analysis of secondary data was conducted.

Data on the numbers of live births and infant deaths were obtained from the published nationwide population register.

Monitoring of IM cases in Kazakhstan is conducted on a daily basis, with the following three levels of information collection and analysis: at the health organization level, the territorial branch level of the Republic Center for Health Development, and the central office of the Republic Center for Health Development.

At the health organization level, data are collected from newborns based on hospital records, or in cases of stillbirth or death of children under one year old. At the territorial branch level of the Republic Center for Health Development, data are collected from all medical organizations, and output forms are generated for each region, including districts and health organizations within them. At the central office of the Republic Center for Health Development, data are collected from the entire republic, and output data are generated at the country level, including regions and districts.

Each health organization designates responsible individuals for monitoring and analyzing incoming data on a daily, monthly, or yearly basis. Trained medical personnel with expertise in examining documents related to live births, stillbirths, and deaths of children under one year old ensure the reliability of the registers and transmit the data to the next level. The compiled data are made available to the public in annual compilation reports entitled “Population Health and Healthcare Organizations’ Performance in the Republic of Kazakhstan” from 2000 to 2020.

The mandatory documentary evidence for infant mortality cases includes the medical certificate of death and the conclusion of perinatal death. These documents must be promptly completed, regardless of where the death occurs (in a maternity hospital, during childhood illness, or at home). The doctor responsible for handling the death observation must possess the necessary medical knowledge and qualifications [46].

IMR data, socioeconomic, health service-related, and population health-related indicators of Kazakhstan were obtained from the annual statistical reports “Population Health and Healthcare Organizations’ Performance in the Republic of Kazakhstan” from 2000 to 2020. These reports contain country- and region-level data on health services and population health. Kazakhstan’s health data collection policy adheres to international guidelines to guarantee statistical data comparability. The IMRs of Kyrgyzstan, Tajikistan, Uzbekistan, and Turkmenistan were derived from World Bank databases [47].

### 2.2. Statistical Analysis

Data were analyzed with IBM SPSS Statistics version 26.0. Linear regression was used for IMR trend analysis in CA countries from 2000 to 2020 and for exploring associations between IMR and socioeconomic factors, health service-related factors, and population health indicators-related factors. Due to the non-normal distribution of the IMR variable, a gamma generalized linear model was used. For the regression, the dependent variable used was the infant mortality rate, which is calculated as the ratio of the number of children who died before the age of 1 year in a given year to the number of live births in the same year, multiplied by 1000. However, to ensure more accurate calculations, the authors adopted the methodology proposed by the German statistician Johannes Raths [48]. Before fitting the regression model, the authors performed a cross-correlation analysis among 69 variables. The regression analyses included 39 indicators, which were further divided into two groups, socioeconomic factors and population health-related factors (Appendix A, Table A1).

## 3. Results

### 3.1. IM Situation in CA Countries and Its Trends

A decline in IMR is observed in all CA countries. The lowest IMR in the CA countries, in 2000 and 2020, was in Kazakhstan and was 36.7 per 1000 live births and 8.9 per 1000 live births, respectively. In 2000, Tajikistan had the highest IMR among CA countries (67.6 per 1000 live births), while in 2020, IMR was highest in Turkmenistan (36.1 per 1000 live births; Figure 1).

In general, in all CA countries, a 20-year decreasing trend of IMR is observed. Thus, a regression analysis of IMR in the CA over 20 years showed that the largest annual decline in IMR was detected in Uzbekistan. The annual value of IMR in Uzbekistan decreased by an average of 1.97 (b = −1.97, intercept = 3979.82; 95% CI: −2.13, −1.79; *p*-value < 0.001), and the infant mortality rate in 2020 decreased by 75% compared to 2000. The annual IMR reduction in Turkmenistan over 20 years was on average 0.94 (b = −0.94, intercept = 1936.6; 95% CI: −1.21, −0.68; *p*-value < 0.001), and IMR in 2020 decreased by 37% compared to 2000, which is the lowest in CA. The annual IMR, decreasing in Kazakhstan over 20 years, was on average 1.49 (b = −1.49, intercept = 3023.44; 95% CI: −1.69, −1.36; *p*-value < 0.001), and IMR in 2020 decreased by 76% compared to 2000.

### 3.2. Description of Socioeconomic and Population Health Indicators-Related Factors

The study included 21 observations collected between 2000 and 2021. Among the socioeconomic factors analyzed, the Gini index, which measures income distribution, ranged from 27.00 to 40.00, with a mean value of 30.04 ± 3.72. The average gross enrollment in tertiary education, calculated as the number of postsecondary students divided by the corresponding population and multiplied by 100, was 52.47 ± 6.49, ranging from 38 to 67. Health expenditure, expressed as total per capita at the average exchange rate, averaged 223.75 ± 105.88, ranging from 51 to 378. GDP ranged from 3.00 to 4.00, with a mean value of 3.22 ± 0.47. The mean income per capita, a measure of economic well-being, was USD 216.93 ± 105.98, ranging from USD 45.00 to 371.00. The proportion of people living below the poverty line ranged from 3.00 to 47.00, with a mean value of 14.44 ± 15.29. The number of births to mothers aged 15–19 ranged from 2.00 to 10.00, with a mean value of 5.08 ± 1.94.

Moving on to population health-related factors, the average bed density per 1000 children was 5.08 ± 1.94, ranging from 3.49 to 5.34. Maternal mortality ratio (MMR), defined as the number of maternal deaths per 100,000 live births within a specific period, averaged 29.50 ± 16.12, ranging from 12 to 61.00. The mortality rate from respiratory system diseases per 1000 live births ranged from 5.00 to 52.90, with a mean value of 18.39 ± 15.19. The average mortality rate from acute upper respiratory infections, influenza, and pneumonia per 1000 live births was 17.66 ± 15.09, ranging from 4.44 to 51.80. Infectious and parasitic diseases had an average mortality rate of 6.72 ± 4.03 per 1000 live births, ranging from 3.07 to 17.00. The mortality rate from intestinal infectious diseases per 1000 live births ranged from 0.10 to 7.70, with a mean value of 1.86 ± 2.09. The mortality rate from congenital malformations per 1000 live births ranged from 15.61 to 38.90, with a mean value of 25.95 and a standard deviation of 6.81. Mortality rate from injury, poisoning, and other external causes per 1000 live births ranged from 2.64 to 10.30, with a mean value of 6.26 ± 2.01. The average malnutrition rate was 26.48 ± 19.26, with a range of 3.30 to 57.80. For the remaining indicators, please refer to Table A1 in Appendix A.

### 3.3. IMR and Its Associated Factors

Regression analysis showed that socioeconomic indicators (Table 1), such as total unemployment (0.163; 0.0001), Gini index (0.725; 0.005), current health expenditure (0.187; 0.0001), GDP (0.187; 0.0001), the proportion of people living below the poverty line (0.072; 0.0001), and babies born to 15–19 years old mothers (0.25; 0.0001) were associated with increased IMR. However, the next indicators from this category individuals using the internet (−0.006; 0.0001), mean age of the population (−1.45; 0.001), total divorce rate (−0.753; 0.0001), gross enrollment in tertiary education (18–22 years) (−0.042; 0.002), health expenditure in USD (−0.003; 0.003), THE (−0.001; 0.001), mean income per capita in USD (−0.002; 0.0001), and the value of the subsistence minimum (−0.008; 0.0001), were associated with decreased IMR.

Additionally, bed density (−0.008; 0.0001) was associated with decreased IMR. All population health-related factors were associated with increased IMR. Thus, MMR (0.009; 0.0001), the infant communicable diseases morbidity rate (0.012; 0.001), intestinal infectious diseases (0.015; 0.0001), sepsis (1.524; 0.001), diseases of the blood and blood-forming organs (0.005; 0.0001), nutritional anemias (0.005; 0.0001), endocrine disorders (0.007; 0.0001), malnutrition (0.018; 0.0001), disorders of the central nervous system (0.012; 0.0001), respiratory diseases (0.001; 0.0001), acute upper respiratory infections (0.001; 0.0001), digestive system diseases (0.013; 0.0001), diseases of the genitourinary system (0.034; 0.0001), certain conditions originating in the perinatal period (0.003; 0.0001), injury, poisoning and certain other consequences of external causes (0.118; 0.0001), mortality rate from diseases of the respiratory system (0.031; 0.0001), mortality rate from acute upper respiratory infections, influenza and pneumonia (0.031; 0.0001), mortality rate from intestinal infectious diseases (0.042; 0.0001), mortality rate from congenital malformations (0.107; 0.012), and mortality rate from injury, poisoning and certain other consequences of external causes (0.148; 0.0001) were associated with increased IMR.

## 4. Discussion

In this study, we analyzed trends in infant mortality rates in CA countries and its determinants in Kazakhstan. We found that all CA countries have successfully decreased their IMR from relatively high levels, with Kazakhstan having the lowest rate in 2000 and 2020. Our results suggest that socioeconomic indicators, such as total unemployment, Gini index, current health expenditure, GDP, the proportion of people living in poverty, and births by 15–19-year-old mothers, were associated with increased infant mortality rates in Kazakhstan, which is consistent with findings from other studies conducted in middle-income countries [49,50,51]. Higher national income and increased public health expenditure have been found to significantly improve health outcomes and reduce infant mortality rates [34,35,52]. Maternal mortality emerges as a notable risk factor for infant mortality, followed by inadequate access to sanitation, water, and lower female education. In Europe, towards the end of the study period, out-of-pocket health expenditure emerged as a significant determinant, deviating from the global trend, which aligns with our research findings [9]. Schell et al. found that income equality (Gini index) was an influential predictor of IMR in middle-income countries, which is consistent with the findings of our study conducted in Kazakhstan. Therefore, extrapolating health policies from high- to low-income countries poses challenges [53]. At the same time, in recent years, Kazakhstan has seen a steady increase in nominal cash income per capita. From 2013 to 2017, this increased by 47.1%, almost 1.5 times. Also, in Kazakhstan, there is a slight decrease in the level of poverty (the share of the population with incomes below the subsistence level) [54]. Our findings imply that poverty levels relate to greater IMR, which is consistent with previous research. Furthermore, regardless of specific maternal socio-demographic, health, and obstetric characteristics, excessive poverty relates to IM [55]. These findings support the idea that major efforts are required to minimize the fraction of individuals living on less than the subsistence level. It is also crucial to consider the influence of government expenditure on healthcare requirements. Increased per capita health spending in high mortality areas is said to play a significant influence in lowering infant mortality. Medical facilities, on the other hand, are more essential in lowering the death rate of children under the age of five [56].

Uzbekistan’s socio-demographic development during the years of independence has resulted in a decline in mortality across all age groups. The country has improved its healthcare system and focused on enhancing maternal and child health [13]. Despite its relatively low GDP level and limited investments in the health sector, Kyrgyzstan has successfully reduced newborn mortality rates by 46% [14].

Access to healthcare resources is another major element in lowering IMR. It has been found that the risk of infant mortality lowers in higher income families, i.e., those with the best access to healthcare services. At the same time, the risk of IM is higher in rural households than in urban ones, which can be explained by the difficulty of obtaining health services in rural regions [57]. In recent years, Kazakhstan implemented various measures to manage IMR. These include the regionalization of prenatal care, establishment of additional perinatal clinics, and equipping them with modern technology. Additionally, training centers have been set up across the country to ensure efficient prenatal care, integrated management of childhood diseases, and effective coordination of perinatal care through the involvement of regional coordinators [11]. The great majority of pregnant women in Kazakhstan received specific prenatal care, according to the 2015 Kazakhstan Multiple Indicator Cluster Survey. Thus, 99.3% of women of reproductive age who gave birth to a live child in the previous two years received care from a competent health provider at least once, and 95.3% received care from a qualified health provider at least four times. Over the last two years, 99.4% of births took place in the presence of trained medical staff and within a medical facility [54].

Rising income and access to safe water are some of the reasons for improved health outcomes in low- and middle-income countries [58]. However, the relative importance of major health determinants varies between income levels, and extrapolating health policies from high-income to low-income countries can be problematic [53]. In our study, we found that indicators such as individuals using the internet, mean age of the population, total divorce rate, gross enrollment in tertiary education, health expenditure in USD, total health expenditure, mean income per capita, and value of the subsistence minimum were associated with decreased IMR in Kazakhstan.

It is important to acknowledge the limitations of our study. One of the main limitations is that we did not take into account the impact of the COVID-19 pandemic on infant mortality rates in Kazakhstan. The pandemic has had direct and indirect effects on infant mortality rates, especially in low- and middle-income countries [39]. Disruptions in antenatal care and follow-up visits due to the pandemic may have contributed to increased rates of stillbirth and infant mortality [40,41]. Future research could examine the impact of the pandemic on infant mortality rates in Kazakhstan. Additionally, maternal factors such as preterm births, birth spacing, multiple/single gestation, type of delivery, and number of abortions were not included in our study and should be considered when analyzing IMR [59,60,61,62].

Sudden infant death syndrome (SIDS) used to be one of the major causes of death of infants [63]. Even though the incidence of SIDS has steadily declined due to adoption of recommendations such as a supine sleeping position, avoiding maternal smoking during pregnancy, and several others, the exact mechanism of the development of this syndrome still remains unclear. The triple-risk model, which was first put forth by Filiano and Kinney, appears to provide the most comprehensive explanation of SIDS pathophysiology [64,65]. While it was out of the scope of our study, discovery of the etiological factors of sudden infant death syndrome would be a significant input towards reducing IM.

It Is worth noting that data quality and accessibility of vital and health statistics are important considerations in IMR. While data quality and accessibility are generally good in Kazakhstan, other countries in the CA region may face some difficulties in this regard [66]. In conclusion, while our study provides valuable insights into the trends and patterns of infant mortality rates in Kazakhstan, it is important to take into account these limitations and factors for future research in this area.

## 5. Conclusions

The decline in infant mortality rates in CA countries over the past two decades is encouraging. However, significant challenges remain in reducing IMR in the region further. This study identified various social determinants of infant health that require attention for improving infant health outcomes in a middle-income country from the region. Improving socioeconomic conditions, investing in healthcare systems, reducing poverty and income inequality, improving access to education, reducing fertility rates, addressing environmental factors, and reducing domestic violence are all potential avenues for further improvement. Furthermore, the study identified several population health and health service-related factors that were associated with increased infant mortality rates. Addressing these factors may be critical for improving maternal and child health outcomes in the region.

## Figures and Tables

**Figure 1 children-10-00923-f001:**
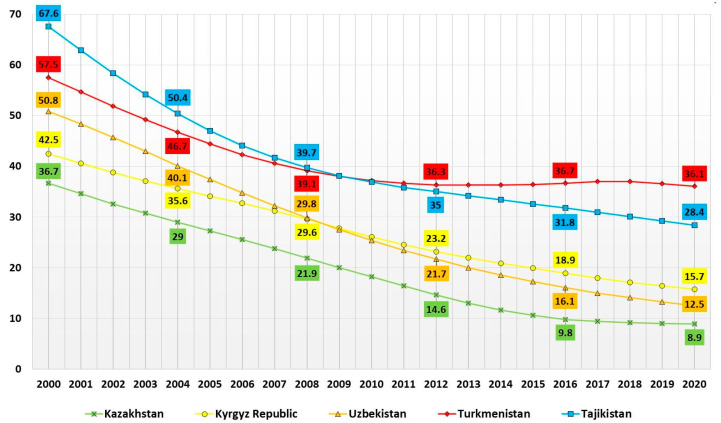
Infant mortality rate in CA countries per 1000 live births, 2000–2020.

**Table 1 children-10-00923-t001:** Regression model coefficient estimates.

Variable		95% Wald Confidence Interval		
	B	Lower	Upper	*p*-Value
Socioeconomic factors				
TU	0.163	0.121	0.206	0.0001
INTERNET	−0.006	−0.007	−0.006	0.0001
G—INDEX	0.725	0.215	1.235	0.005
AGE	−1.45	−2.47	−0.43	0.005
TDR	−0.753	−0.841	−0.664	0.0001
GENROLL	−0.042	−0.07	−0.015	0.002
CHE	0.187	0.095	0.28	0.0001
HE USD	−0.003	−0.005	−0.001	0.003
THE KZT	−0.001	−0.002	0	0.001
GDP KZT	0.187	0.095	0.28	0.0001
INCOME USD	−0.002	−0.003	−0.002	0.0001
POVERTY	0.072	0.057	0.086	0.0001
SUBMIN USD	−0.008	−0.01	−0.006	0.0001
YOUNG MOTHER	0.25	0.142	0.358	0.0001
Population health-related factors				
BED	−0.008	0.574	0.891	0.0001
MMR	0.009	0006	0.013	0.0001
MORBIDITY	0.0004	0.0004	0.001	0.0001
COMD	0.012	0.01	0.013	0.0001
INTESTID	0.015	0.013	0.017	0.0001
SEPSIS	1.524	0.6	2.448	0.001
BLOOD	0.005	0.004	0.007	0.0001
ANAEMIA	0.005	0.004	0.006	0.0001
ENDOCRINE	0.007	0.007	0.008	0.0001
MALNUTRITION	0.018	0.017	0.019	0.0001
CNERVE	0.012	0.01	0.015	0.0001
NERVE	0.011	0.01	0.011	0.0001
RESP	0.001	0.001	0.001	0.0001
ARESP	0.001	0.001	0.001	0.0001
DIGESTIVES	0.013	0.012	0.014	0.0001
GENITS	0.034	0.023	0.045	0.0001
CERTAIN	0.003	0.003	0.003	0.0001
INJURY	0.118	0.085	0.151	0.0001
MRRESP	0.031	0.026	0.035	0.0001
/MRARESP	0.031	0.026	0.035	0.0001
MRINFECT	0.331	0.239	0.422	0.0001
MRINFECT	0.42	0.209	0.63	0.0001
MRMALFORMATIONS	0.107	0.024	0.19	0.012
MRINJURY	0.148	0.122	0.173	0.0001

## Data Availability

Not applicable.

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
