# Peer review of "Infant Mortality Trends and Determinants in Kazakhstan"

_children, 2023, doi:10.3390/children10060923_

Round 1

Reviewer 1 Report

Well-done on your study. Please see below my comment on your study.

1) Since your study is on Kazakhstan and CA, I would have expected that your buildup to the study rationale will be focussed more on Kazakhstan and CA. Your entire introduction was on other countries with only the last paragraph briefly mentioning Kazakhstan. If this study is on Kazakhstan and CA, then include more literature review on infant mortality in Kazakhstan and CA to make for a stronger study rationale.

2) The material and methods section is grossly insufficient. Please provide more details on the data source. Provide more details on the data collection methodology. What are the dependent variables? which are the descriptive Study Variables?

3) How were the dependent variables expressed?

4) Report both the adjusted and Unadjusted odds ratio with confidence intervals.

5) A discussion section is essentially non-existent in your study. Please provide a discussion section that comprehensively and specifically discusses the results of the study either in agreement with or opposing previous studies.

6) Overall, more focus need to be given to Kazakhstan in the literature as it is the country being analysed.

English is fine just require minor checks.

Author Response

Response to Reviewers

For your convenience we also upload the MS Word file with the response

The authors thank the reviewers for the time and effort spent to improve the manuscript. All the comments are really appreciated. We agree in general with the majority of points raised by the reviewers and have revised the manuscript considering the suggestions. All of them were very insightful and inspirational.

We hope that the revised manuscript may be contributing to the journal’s interests and values.

The text below follows all the comments from the reviewers. The reviewer´s text is in black and authors´ replies are in red (same as updates and changes in the revised manuscript).

Response to Reviewer 1’s comments:

1) Since your study is on Kazakhstan and CA, I would have expected that your buildup to the study rationale will be focussed more on Kazakhstan and CA. Your entire introduction was on other countries with only the last paragraph briefly mentioning Kazakhstan. If this study is on Kazakhstan and CA, then include more literature review on infant mortality in Kazakhstan and CA to make for a stronger study rationale.

Thank you for your valuable note. We added a paragraph about studies on Kazakhstan and CA. Overall, your comments helped us to realize what we missed, thank you for that. The revised introduction part is stated as following:

“Kazakhstan has witnessed a notable reduction in mortality rates among all age groups, including infants, since 2000 [5]. Nevertheless, despite considerable investments in the health sector, Kazakhstan continues to fall behind OECD countries when it comes to key health indicators. The population of the country, which faced a decline between 1992 and 2002, has witnessed a subsequent growth of 20%. As of 2019, Kazakhstan's Gross Domestic Product (GDP) was estimated at USD 180.2 billion, with a per capita gross national income of USD 8810. These figures are significantly lower compared to the USD 40,115 observed in other OECD countries [6]. According to time-series analysis conducted by Thomas, the transition to a market economy in the 1990s temporarily alleviated environmental degradation but also resulted in socio-economic challenges and a decline in living standards [7]. Some credibility issues have been raised regarding infant mortality data, including underreporting/misreporting in the Central Asian region and Eastern European countries [8,9]. However, in 2021, Kazakhstan, Kyrgyzstan, Uzbekistan, and Tajikistan were among the 36 countries with high-quality national data used for child mortality estimation by UN experts [5]. Furthermore, Kazakhstan implemented international criteria for live births and stillbirths in 2008, leading to increased attention and resources for the care of low-weight newborns [10]. Over the past decade, mortality rates in Kazakhstan have declined across all age groups, including infants [11]. Regions such as Kostanay, Akmola, and Almaty have observed relatively high neonatal mortality rates [12]. In the Central Asian countries, neonatal mortality has experienced a 66% decline between 1990 and 2021, with an average annual reduction of 3.4% [5].

Uzbekistan has also achieved a decline in the death toll across all age groups, thanks to improvements in the healthcare system and an enhanced focus on maternal and child health [13]. Despite its relatively low GDP level, Kyrgyzstan has made significant progress in reducing infant mortality, which was successfully reduced by 46% over the last two decades [14].”

2) The material and methods section is grossly insufficient. Please provide more details on the data source. Provide more details on the data collection methodology.

Thank you for your suggestion. We added paragraph about data source in the methods part. The revised method part is stated as following:

“Monitoring of IM cases in Kazakhstan is conducted on a daily basis, with three levels of information collection and analysis: at the health organization level, the territorial branch level of the Republic Center for Health Development, and the central office of the Republic Center for Health Development.

At the health organization level, data is collected from newborns based on hospital records or in cases of stillbirth or death of children under one year old. At the territorial branch level of the Republic Center for Health Development, data is collected from all medical organizations, and output forms are generated for each region, including districts and health organizations within them. At the central office of the Republic Center for Health Development, data is collected from the entire republic, and output data is generated at the country level, including regions and districts.

Each health organization designates responsible individuals for monitoring and analyzing incoming data on a daily, monthly, or yearly basis. Trained medical personnel with expertise in examining documents related to live births, stillbirths, and deaths of children under one year old ensure the reliability of the registers and transmit the data to the next level. The compiled data is made available to the public in the annual compilation report titled "Population Health and Healthcare Organizations' Performance in the Republic of Kazakhstan from 2000 to 2020."

The mandatory documentary evidence for infant mortality cases includes the medical certificate of death and the conclusion of perinatal death. These documents must be promptly completed regardless of where the death occurs (in a maternity hospital, during childhood illness, or at home). The doctor responsible for handling the death observation must possess the necessary medical knowledge and qualifications [34].”

What are the dependent variables?

In regression, the dependent variable used is the Infant Mortality Rate (IMR), which is calculated as the ratio of the number of children who died before the age of 1 year in a given year to the number of live births in the same year, multiplied by 1,000. However, to ensure more accurate calculations, the authors adopted the methodology proposed by the German statistician Johannes Raths

Reference: [Order of the Chairman of the Committee on Statistics of the Ministry of National Economy of the Republic of Kazakhstan dated September 21, 2018, â„– 1. Registered in the Ministry of Justice of the Republic of Kazakhstan on October 16, 2018, â„– 17555, https://adilet.zan.kz/rus/docs/V1800017555].

Which are the descriptive Study Variables?

In the Results section we added paragraph 3.2. Description of socioeconomic and population health indicators-related factors and Annex Table A3. The revised Results part is stated as following:

“The study included 21 observations collected between 2000 and 2021. Among the socio-economic factors analyzed, the Gini index, which measures income distribution, ranged from 27.00 to 40.00, with a mean value of 30.04 ± 3.72. The average gross enrollment in tertiary education, calculated as the number of postsecondary students divided by the corresponding population and multiplied by 100, was 52.47 ± 6.49, ranging from 38 to 67. Health expenditure, expressed as total per capita at the average exchange rate, averaged at 223.75 ± 105.88, ranging from 51 to 378. The GDP variable ranged from 3.00 to 4.00, with a mean value of 3.22 ± 0.47. The mean income per capita, a measure of economic well-being, was 216.93 ± 105.98 in US dollars, ranging from 45.00 to 371.00. The proportion of people living below the poverty line ranged from 3.00 to 47.00, with a mean value of 14.44 ± 15.29. The number of births to mothers aged 15-19 ranged from 2.00 to 10.00, with a mean value of 5.08 ± 1.94.

Moving on to population health-related factors, the average bed density per 1000 children was 5.08 ± 1.94, ranging from 3.49 to 5.34. Maternal mortality ratio (MMR), defined as the number of maternal deaths per 100,000 live births within a specific period, had an average of 29.50 ± 16.12, ranging from 12 to 61.00. The mortality rate from respiratory system diseases per 1000 live births ranged from 5.00 to 52.90, with a mean value of 18.39 ± 15.19. The average mortality rate from acute upper respiratory infections, influenza, and pneumonia per 1000 live births was 17.66 ± 15.09, ranging from 4.44 to 51.80. Infectious and parasitic diseases had an average mortality rate of 6.72 ± 4.03 per 1000 live births, ranging from 3.07 to 17.00. Mortality rate from intestinal infectious diseases per 1000 live births ranged from 0.10 to 7.70, with a mean value of 1.86 ± 2.09. The mortality rate from congenital malformations per 1000 live births ranged from 15.61 to 38.90, with a mean value of 25.95 and a standard deviation of 6.81. Mortality rate from injury, poisoning, and other external causes per 1000 live births ranged from 2.64 to 10.30, with a mean value of 6.26 and a standard deviation of 2.01. The average malnutrition rate was 26.48 ± 19.26, with a range of 3.30 to 57.80. For the remaining indicators, please refer to Table A3 in Appendix A.”

3) How were the dependent variables expressed?

We have added sentences about dependent variable in Material and Methods section (2.2. Statistical Analysis) and the text was changed as following:

“In regression, the dependent variable is used as IMR, which is the ratio of the number of children who died before the age of 1 year in a given year to the number of live births in a given year, multiplied by 1,000. However, for greater accuracy of calculation, the methodology proposed at the beginning of the century by the German statistician Johannes Raths and named after him is used.” [Order of the Chairman of the Committee on Statistics of the Ministry of National Economy of the Republic of Kazakhstan dated September 21, 2018, â„– 1. Registered in the Ministry of Justice of the Republic of Kazakhstan on October 16, 2018, â„– 17555, https://adilet.zan.kz/rus/docs/V1800017555].

4) Report both the adjusted and Unadjusted odds ratio with confidence intervals.

Thank you for the very valuable comment.

We considered finding the adjusted and unadjusted odds ratios, however what we found out (also in literature) is that the adjusted and unadjusted odds ratios are calculated only when fitting logistic regression and are not possible in this study. One of the goals of our study is to identify the determinants of infant mortality in Kazakhstan using linear regression, not to predict it. Hopefully you can accept our comment.

Reference:

Weaver, Bruce. (2018). Re: Is there a manner to obtain an Odds ratio or relative risk by using linear regression?. Retrieved from: https://www.researchgate.net/post/Is-there-a-manner-to-obtain-an-Odds-ratio-or-relative-risk-by-using-linear-regression/5b97a7018b950062c629dd14/citation/download.

5) A discussion section is essentially non-existent in your study. Please provide a discussion section that comprehensively and specifically discusses the results of the study either in agreement with or opposing previous studies.

Dear reviewer, we have comprehensively updated the discussion section. We did not insert the updates here, we kindly ask to check in the manuscript text (highlighted with red text colour)

6) Overall, more focus need to be given to Kazakhstan in the literature as it is the country being analysed.

We totally agreed and attempted to add more about Kazakhstan from the literature. All changes displayed at the introduction and discussion parts of the manuscript.

Reviewer 2 Report

This study used infant mortality to study socioeconomic factors in nine Central Asian countries and Kazakhstan (a middle-income country) using linear regression from 2000 to 2020. The study explored associations between infant mortality rate and socioeconomic factors, health service-related factors, and population health indicators-related factors. The analysis revealed that infant mortality rates decreased in all CA countries, with Kazakhstan having the lowest rate in 2000 and 2020. The results suggested that socioeconomic indicators, such as total unemployment, Gini index, current health expenditure, GDP, the proportion of people living below the poverty line, and births by 15-19-year-old mothers, were associated with increased infant mortality rates. Improving socioeconomic conditions, investing in healthcare systems, reducing poverty and income inequality, and improving access to education, represent challenges for the future.

The paper is generally well-written, however, the abstract could be more concise.

This study used infant mortality to study socioeconomic factors in nine Central Asian countries and Kazakhstan (a middle-income country) using linear regression from 2000 to 2020. The study explored associations between infant mortality rate and socioeconomic factors, health service-related factors, and population health indicators-related factors. The analysis revealed that infant mortality rates decreased in all CA countries, with Kazakhstan having the lowest rate in 2000 and 2020. The results suggested that socioeconomic indicators, such as total unemployment, Gini index, current health expenditure, GDP, the proportion of people living below the poverty line, and births by 15-19-year-old mothers, were associated with increased infant mortality rates. Improving socioeconomic conditions, investing in healthcare systems, reducing poverty and income inequality, and improving access to education, represent challenges for the future.

The paper is generally well-written, however, the abstract could be more concise.

Author Response

Response to Reviewer 2’s comments:

For your convenience please check the MS Word file we uploaded

This study used infant mortality to study socioeconomic factors in nine Central Asian countries and Kazakhstan (a middle-income country) using linear regression from 2000 to 2020. The study explored associations between infant mortality rate and socioeconomic factors, health service-related factors, and population health indicators-related factors. The analysis revealed that infant mortality rates decreased in all CA countries, with Kazakhstan having the lowest rate in 2000 and 2020. The results suggested that socioeconomic indicators, such as total unemployment, Gini index, current health expenditure, GDP, the proportion of people living below the poverty line, and births by 15-19-year-old mothers, were associated with increased infant mortality rates. Improving socioeconomic conditions, investing in healthcare systems, reducing poverty and income inequality, and improving access to education, represent challenges for the future.

The paper is generally well-written, however, the abstract could be more concise.

Thank you very much for your valuable feedback and inspirational conclusion of your review. We hope you will let us have the current version of the abstract which seems already concise, however we made slight changes. We hope our abstract effectively captures the essence of our research.

Reviewer 3 Report

The authors in the abstract and in the text use the abbreviation GDP without describing the acronym, in the abstract they speak of Gini index without specifying it, the title presents a generic country of Asia without indicating the area. The mortality of the first five years is not distinguished, according to the sudden deaths Still birth just mentioned, they do not mention suid and sids (sudden deaths of the first year of life). They speak generically of environmental causes, but smoking in pregnancy and parental is very important, they describe respiratory deaths without making a mention. They do not mention a real etiopathogenesis. It looks like a list. The category of the first 5 years of life must be divided from the first year of life because the causes of death are differently important, also from a social point of view.

They do not mention Filiano and Kinney's 1994 triple risk of death theory. The description of the covid cause is very generic

Author Response

Response to Reviewers

For your convenience please check the MS Word file with the response 

The authors thank the reviewers for the time and effort spent to improve the manuscript. All the comments are really appreciated. We agree in general with the majority of points raised by the reviewers and have revised the manuscript considering the suggestions. All of them were very insightful and inspirational.

We hope that the revised manuscript may be contributing to the journal’s interests and values.

The text below follows all the comments from the reviewers. The reviewer´s text is in black and authors´ replies are in red (same as updates and changes in the revised manuscript).

Response to Reviewer 3’s comments:

The authors in the abstract and in the text use the abbreviation GDP without describing the acronym,

Thank you very much for your remark. In the abstract we wrote out the acronym GDP as gross domestic product.

in the abstract they speak of Gini index without specifying it,

We agree with your comment. We decided to leave as it is in abstract, because one of the reviewers insisted to have our abstract concise, however we attempted to describe Gini index ( a measure of the distribution of income across a population) it in the introduction section:

“The Gini index, which measures income distribution across a population, indicates the level of inequality present. A higher Gini index suggests greater inequality [31]. There was a positive and significant correlation observed between infant mortality and the Gini index [29].”

the title presents a generic country of Asia without indicating the area.

We appreciate the idea and we changed the name of the article as “Infant mortality trends and determinants in Kazakhstan”.

The mortality of the first five years is not distinguished, according to the sudden deaths Still birth just mentioned, they do not mention suid and sids (sudden deaths of the first year of life).

We are really thankful for your comment. We understand that it is our miss and we mentioned it as one of limitations in the discussion part:

“We did not cover sudden infant death syndrome (SIDS) used to be one of the major causes of death of infants [64]. Even though the incidence of SIDS demonstrated a steady decline due to adoption of the recommendations such as supine sleeping position, avoiding maternal smoking during pregnancy and several others, the exact mechanism of the development of that syndrome still remains unclear.”

They speak generically of environmental causes, but smoking in pregnancy and parental is very important, they describe respiratory deaths without making a mention.

They do not mention a real etiopathogenesis. It looks like a list.

You fairly pointed out what we lack. We attempted to update our results part and also added the Annex tables. Please, accept our current attempt as it is, hopefully we can improve as we gain more data and experience in future.

The category of the first 5 years of life must be divided from the first year of life because the causes of death are differently important, also from a social point of view.

We totally agree with you. Sorry, we did not cover it comprehensively. In our study we tried to focus only on infant mortality. The broader age group was out of our scope. However, your comments are really insightful, so we will apply them in the next study attempts.

They do not mention Filiano and Kinney's 1994 triple risk of death theory.

We are really thankful for your comment. We mentioned its importance in the discussion part:

“We did not cover sudden infant death syndrome (SIDS) used to be one of the major causes of death of infants [64]. Even though the incidence of SIDS demonstrated a steady decline due to adoption of the recommendations such as supine sleeping position, avoiding maternal smoking during pregnancy and several others, the exact mechanism of the development of that syndrome still remains unclear. The triple-risk model, which was first put forth by Filiano and Kinney, appears to provide the most comprehensive explanation of SIDS pathophysiology [65, 66]. While it was out of the scope of our study, discovery of the etiological factors of the sudden infant death syndrome would be a significant input towards declining IM.”

The description of the covid cause is very generic

We appreciate your valuable feedback and suggestions. After carefully considering your comments, we acknowledge that the description of the COVID-19 cause in our paper may have been generic. We apologise for any confusion caused. In our limitations part of the paper, we mention that it is our limitation. Hopefully, in future with more data from our region we can address all the concerns.

Thank you once again for your time and attention to our paper. We look forward to your continued support and feedback throughout this process.

Round 2

Reviewer 3 Report

Well done!

Know the article is justly corrected to be published